# Immunological Mechanisms of Vaccine-Induced Protection against SARS-CoV-2 in Humans

**Keshav Goyal** [1], **Harsh Goel** [2], **Pritika Baranwal** [1], **Anisha Tewary** [1], **Aman Dixit** [1], **Avanish Kumar Pandey** [2], **Mercilena Benjamin** [2], **Pranay Tanwar** [2], **Abhijit Dey** [3], **Fahad Khan** [4], **Pratibha Pandey** [4], **Piyush Kumar Gupta** [5], **Dhruv Kumar** [6], **Shubhadeep Roychoudhury** [7], **Niraj Kumar Jha** [8], **Tarun Kumar Upadhyay** [9,*] and **Kavindra Kumar Kesari** [10,*]

1 Department of Microbiology, Ram Lal Anand College, University of Delhi, New Delhi 110019, India; keshavgoyal252@gmail.com (K.G.); baranwalpritika@gmail.com (P.B.); anishatewary5@gmail.com (A.T.); aman.devdixit564@gmail.com (A.D.)

2 Department of Laboratory Oncology, All India Institute of Medical Sciences, New Delhi 110029, India; goel.harsh271@gmail.com (H.G.); avikpandey2726@gmail.com (A.K.P.); marslena@gmail.com (M.B.); pranaytanwar@gmail.com (P.T.)

3 Department of Life Sciences, Presidency University, College Street, Kolkata 700073, India; abhijit.dbs@presiuniv.ac

4 Department of Biotechnology, Noida Institute of Engineering & Technology, 19, Knowledge Park-II, Institutional Area, Greater Noida 201306, India; fahadintegralian@gmail.com (F.K.); shukla.pratibha1985@gmail.com (P.P.)

5 Department of Life Sciences, School of Basic Sciences and Research, Sharda University, Plot No. 32-34, Knowledge Park III, Greater Noida 201310, India; piyush.kumar1@sharda.ac.in

6 Amity Institute of Molecular and Stem Cell Research (AIMMSCR), Amity University Uttar Pradesh, Noida 201301, India; dkumar13@amity.edu

7 Department of Life Science and Bioinformatics, Assam University, Silchar 788011, India; shubhadeep1@gmail.com

8 Department of Biotechnology, School of Engineering and Technology (SET), Sharda University, Greater Noida 201310, India; niraj.jha@sharda.ac.in

9 Animal Cell Culture and Immuno-Biochemistry Lab, Department of Biotechnology, Parul Institute of Applied Sciences and Centre of Research for Development, Parul University, Vadodara 391760, India

10 Department of Applied Physics, School of Science, Aalto University, 00076 Espoo, Finland

* Correspondence: tarun_bioinfo@yahoo.co.in or tarunkumar.upadhyay18551@paruluniversity.ac.in (T.K.U.); kavindra.kesari@aalto.fi (K.K.K.)

**Abstract:** The SARS-CoV-2 infection spread rapidly throughout the world and appears to involve in both humoral and cell-mediated immunity. SARS-CoV-2 is attached to host cells via binding to the viral spike (S) proteins and its cellular receptors angiotensin-converting enzyme 2 (ACE2). Consequently, the S protein is primed with serine proteases TMPRSS2 and TMPRSS4, which facilitate the fusion of viral and cellular membranes result in the entry of viral RNA into the host cell. Vaccines are urgently required to combat the coronavirus disease 2019 (COVID-19) outbreak and aid in the recovery to pre-pandemic levels of normality. The long-term protective immunity is provided by the vaccine antigen (or pathogen)-specific immune effectors and the activation of immune memory cells that can be efficiently and rapidly reactivated upon pathogen exposure. Research efforts aimed towards the design and development of vaccines for SARS-CoV-2 are increasing. Numerous coronavirus disease 2019 (COVID-19) vaccines have passed late-stage clinical investigations with promising outcomes. This review focuses on the present state and future prospects of COVID-19 vaccines research and development, with a particular emphasis on immunological mechanisms of various COVID-19vaccines such as adenoviral vector-based vaccines, mRNA vaccines, and DNA vaccines that elicits immunological responses against SARS-CoV-2 infections in humans.

**Keywords:** SARS-CoV-2; DNA vaccine; mRNA vaccine; adenoviral vector-based vaccine

## 1. Introduction

The Severe Acute Respiratory Syndrome Coronavirus 2 (SARS-CoV-2) triggered the coronavirus illness 2019 (COVID-19) outbreak in the Chinese city of Wuhan and spread quickly throughout the world. Coronaviruses (CoVs) have emerged as the primary pathogens which increasing several respiratory ailments. CoVs are members of the *Coronaviridae* family, which is part of the *Nidovirales* order. This novel virus appears to be very infectious and dangerous, with an incubation period ranging from 2 to 14 days [1]. It is transmitted mainly by inhalation or contact with contaminated droplets. Given the scale of the COVID 19 issue repertory, the whole power of global research resources should be concentrated toward the need for a viable vaccination [2,3]. Conventional vaccinations have effectively reduced the burden of a variety of infectious illnesses in the past. However, in epidemic circumstances, conventional techniques may not always be appropriate or even practicable. In terms of producibility, outbreak circumstances may hinder traditional vaccine development. One of the critical issues for pandemic preparedness is the unpredictability of new diseases. Because vaccine targets are unknown before an outbreak occurs, time remains a key impediment to successful vaccine development [4,5]. In India, two vaccines have been authorized one is Covishield and another one is Covaxin. Covishield, also known as AZD-1222, is a SARS-CoV-2 spike protein-based vaccine that indicates a 70.42% reduction in COVID-19 event among the vaccinated group, this vaccine is manufactured by the Serum Institute of India in Pune, India, whilst Covaxin is an inactivated vaccine of SARS-CoV-2 exhibits 60% lowering the risk of developing COVID-19 among the vaccinated group after two doses, this vaccine is manufactured by Bharat Biotech in Hyderabad, India. Covaxin is an inactivated vaccine of SARS-CoV-2 shows a 60% lowering risk of developing COVID-19 among the vaccinated group after two doses. Table 1 compares the differences between these two authorized vaccinations in India.

**Table 1.** Shows Comparison between Covaxin and Covishield.

| | Covishield | Covaxin |
|---|---|---|
| DEVELOPED BY | Serum Institute of India | Bharat Biotech ICMR |
| VACCINE TYPE | Non-Replicating Viral Vector | Inactivated |
| EFFICACY | Drugs Controller General of India (DCGI): 70.42% overall | 60% |
| STORAGE TEMPERATURE | 2–8 degree Celsius | 2–8 degree Celsius |
| DOSES | Two Doses (0, 84 Days) | Two Doses (0, 14 Days) |

Virus-based vaccines, nucleic acid-based vaccines, attenuated vaccines, and protein-based vaccines display significant results in clinical trial studies. Currently, 11 novel vaccinations are being tested throughout the world including Pfizer/BioNtech, Moderna, Oxford/AstraZeneca, Bharat Biotech, Sputnik V, SinoVec, Sinopharm, CanSino, Johnson & Johnson, Novavex, and EpiVecCorona, etc. In addition to these, 183 vaccinations are now in preclinical development, with 97 in clinical trials [1]. As of April 2021,28of such vaccines had entered phase III clinical trials, showing effectiveness in peer-reviewed literature, and comprehensive publicly available reports filed by regulatory authorities, leading to emergency approvals for their use in a significant number of countries. Various vaccines, comprising mRNA, adenoviral-vectored, protein subunit, and whole-cell inactivated virus vaccines, have shown their effectiveness in phase III clinical studies and have been granted emergency approval in several countries. A successful COVID-19 vaccine would very certainly require both neutralizing antibodies and a Th1-driven cellular component. Concerns of increased illness from possibly immunopathologic Th2 responses, as shown in animal trials of previous coronavirus vaccines, stimulate the study of immune response induction following vaccination. High-quality functional antibody responses and Th1-biased T-cell responses must be generated to decrease the risk of vaccine-associated enhanced respiratory disease or antibody-dependent amplification of replication. Vaccines that produce high neutralizing antibodies, Th1 responses, and balanced CD4/CD8 and polyfunctional

T cell responses are less likely to cause immunopathology [6–8]. The majority of COVID-19 vaccines are intended to induce immunological responses, preferably neutralizing antibodies (NAbs), against the SARS-CoV-2 spike protein as shown in Table 2. These include mRNA vaccines BNT162b2 which is developed by Pfizer/BioNtech elicits an immunological response including IgG, IgA, CD8+ cells, or CD4+ cells, while mRNA-1273 vaccine developed by Moderna mainly induces CD8 T cell response. Adenoviral-vectored vaccines also induce a variety of immunological responses. ChAdOx1 nCoV-19 vaccine developed by Oxford/AstraZeneca produces anti-IgA and IgG antibodies, T cell, Th1-biased T-cell, IFN-γ and IL-2, CD4+ T cell response, Gam-COVID-Vac (Sputnik V) developed by Gamaleya Research Institute mounts IgG cell response, Ad26.COV2.S (Janssen) developed by Johnson and Johnson induces Th1-biased, Th2-skewed, CD8+ T-cell, IFN-γ, IL-4, IL-5, or IL-10 response. All DNA vaccines mostly evoke T-cell responses. INO-4800 instigates IgG cell response. AG0301-COVID19 provokes neutralizing antibody response. GX-19 elicits Th1-biased T cell responses, CD4+ and CD8+ T cell response. BacTRL-Spike produces both cellular and humoral immunity against spike protein. One protein subunit vaccine (NVX-CoV2372; Novavax, Gaithersburg, MD, USA) and one whole-cell inactivated viral vaccine (BBV152; Bharat Biotech, Hyderabad, India) have both claimed positive effectiveness findings via official corporate press releases, and BBV152 has been granted urgent approval in many countries [9]. Thus, in this review, we describe the immunological mechanisms of various COVID-19 vaccines that elicits immunological responses against SARS-CoV-2 infections in humans.

**Table 2.** List of COVID-19 Vaccines.

| Vaccine Developed by | Name of Vaccine | Mode | Type of Response |
| --- | --- | --- | --- |
| Pfizer/BioNtech | BNT162b2 | mRNA vaccine | IgG, IgA, CD8+ cells or CD4+ cells |
| Moderna | mRNA-1273 | mRNA vaccine | CD8+ T cell |
| CureVac AG | CVnCoV | mRNA vaccine | IL-6, IFN-α |
| Abogen | ARCoV | mRNA vaccine | Th-1 biased |
| Arcturus | ARCT-021 | mRNA vaccine | CD8+ cell-mediated and Th1/Th2-mediated immunity |
| Symvivo | BacTRL-Spike | DNA vaccine | Induce both cellular and humoral immunity against spike protein |
| Genexine | GX-19 | DNA vaccine | Th1-biased T cell responses, CD4+ and CD8+ T cell |
| Inovio | INO-4800 | DNA vaccine | IgG cell |
| AnGes Inc. and Osaka University | AG0301-COVID19 | DNA vaccine | Neutralizing antibodies and T-cell responses |
| Inovio | GLS-5300 | DNA vaccine | T-cell responses, S1-ELISA |
| Oxford/AstraZeneca | ChAdOx1 nCoV-19 | Adenoviral-vectored | Anti-IgA and IgG antibodies, T cell, Th1-biased T-cell, IFN-γ and IL-2, CD4+ T cells |
| Gamaleya Research Institute | Gam-COVID-Vac (Sputnik V) | Adenoviral-vectored | IgG cell |
| Johnson and Johnson | Janssen, Ad26.COV2.S | Adenoviral-vectored | Th1-biased, Th2-skewed, CD8+ T-cell, IFN-γ, IL-4, IL-5, or IL-10 |
| Bharat Biotech | BBV152 | Whole cell inactivated viral vaccine | Th-1 cells, IgG cells |
| Sinovac Biotech | SinoVec | Inactivated-virus COVID-19 Vaccine | T cells |

Table 2. *Cont.*

| Vaccine Developed by | Name of Vaccine | Mode | Type of Response |
|---|---|---|---|
| Beijing Bio-Institute of Biological Products Co Ltd. | Sinopharm | Inactivated-virus COVID-19 Vaccine | Neutralizing antibody GMT, Humoral responses |
| CanSino Biologics Inc. | CanSino | Inactivated-virus COVID-19 vaccine | Specific ELISA antibody responses to the receptor binding domain (RBD) and neutralizing antibody responses |
| Novavex | NVX-CoV2372 | Protein subunit vaccine | CD4+ T-cell, IgG cells |
| Vektor State Research Center of Virology and Biotechnology in Russia | EpiVecCorona | Protein subunit vaccine | CD4+ T-cell |

## 2. Sensing of SARS-CoV-2 Pathogen by Innate Immunity

As an initial line of protection, vertebrates use innate immunological responses as their defensive systems. The innate immune response performs three primary functions: (i) limiting viral replication within infected cells, (ii) inducing an antiviral state in the immediate tissue environment, along with the recruitment of innate immune effector cells, and (iii) priming the adaptive immune response [10,11]. Pathogen recognition receptors (PRRs) distinguish microbial components known as pathogen-associated molecular patterns that are likely to be required for the survival of the microorganism. PRRs are expressed constitutively in the host and recognize infections or their intermediate products throughout the pathogen's life cycle. PRRs react with distinct microbial components different from the self and activate specific signaling pathways that drive biological responses against pathogens [12,13]. In the case of RNA viruses, the mammalian host's innate immune response has two primary routes for controlling viral infections. One includes signaling pathways mediated by TLR 3, 7 or 9 in response to detection of the viral DNA or its intermediates. The other is carried out by cytoplasmic RNA helicases such as melanoma differentiation-associated gene 5 (MDA5) or retinoic acid-inducible gene I (RIG I), which may detect 5′ triphosphorylated and double viral RNAs as well as stimulate the host antiviral innate immune system [14,15]. Several observations highlight the critical role of innate immunity in SARS-CoV-2 regulation. Human coronaviruses such as SARS, MERS, and SARS-CoV-2 have acquired specific immune-suppressive mechanisms such as a part of the viruses' genetic material is committed to code for the proteins that particularly target human innate immunity systems, prominently IFN response pathways by blocking TLR3/7 signaling [16,17]. The innate immune system also triggers activates the adaptive immune responses, which act together just to eradicate infections [11]. However, adaptive immune responses are more specific and provide long-term protection from reinfection with the same type of pathogen [10].

## 3. Humoral and Cell Mediated Immune Responses against SARS-CoV-2

Human SARS-CoV-2 infection appears to include both humoral as well as cell-mediated immunity [18,19]. Antibody molecules produced by plasma cells mediate the humoral immune response. When antigen attaches to the B-cell antigen receptor, it alerts B cells while also being internalized and processed into peptides that activate the armed helper [20]. In cell-mediated immune responses, the second kind of adaptive immune response activated T cells directly respond to a foreign antigen presented to them on the surface of a host cell. For example, a virus-infected host cell bearing viral antigens on its surface may be destroyed by a T cell, eliminating the infected cell before the virus has a chance to reproduce. In other cases, the T cell secretes signal molecules that stimulate macrophages to destroy the intruders they have phagocytized [10]. Adaptive immunity involves the coordination of T and B cell immune responses to the SARS-CoV-2 virus. Immune responses to the SARS virus begin within 7–10 days after infection [21]. Early responses to COVID-19 infection

include IgM and IgA, although it is unclear whether they can alter the course of the illness [22–24]. First, early IgG responses emerge from germinal centers after T follicular cells stimulate naïve B cells, which develop into activated B cells, which then differentiate into B memory cells and IgG generating plasmablasts. B memory cells and long-lived plasma cells in the bone marrow can reactivate antigen-specific responses against SARS-CoV-2 pathogen if exposed again. Moreover, this overlooks the importance of T cell memory for COVID-19 antigenic determinates, which can lead to efficient cytotoxic T cell immunity and aid in B cell responses [25].

## 4. Vaccine-Induced Immune Responses against SARS-CoV-2 Infections

### *4.1. Nucleic Acid-Based Vaccines for COVID-19*

The usage of nucleic acid-based vaccines is a new method of vaccination that induces immune responses comparable to those elicited by live, attenuated vaccines. When nucleic acid vaccines are administered, the endogenous production of viral proteins having native conformation, glycosylation patterns, and other posttranslational modifications that match antigen generated during normal viral infection occurs. To date various protein antigens, nucleic acid vaccines have been demonstrated to elicit both antibody and cytotoxic T-lymphocyte responses. The convenience of the vector, ease of administration, the longevity of expression, and evidence of integration are the advantages of nucleic acid-based vaccines. More research is needed to determine the practicality, safety, and efficacy of this novel and promising technique [26]. Nucleic acid vaccines, at their most basic, use the host's transcriptional and translational machinery to create the target gene product. This polypeptide product is then identified by immune system components. The initial research focused on the absorption of plasmid DNA by myocytes. Despite the fact that myocytes can deliver antigen to immune cells, they are not the major activators of immune cells. An immune response is instead largely initiated by antigen presentation cells (APC) include dendritic cells (DC), B cells, macrophages, and Langerhans cells [27,28]. The most potent APC, dendritic cells acquire antigen via three major mechanisms. First, nucleic acid vaccines may be directly transfected into DC. Second, DC takes up soluble antigen produced or released by transfected cells from interstitial areas. Third, and probably most intriguingly, DC preferentially takes up damaged or dead cells as a result of the vaccination or its action [29].

### 4.1.1. Immunological Mechanisms of Different m-RNA Vaccine-Induced Protection against SARS-CoV-2 RNA Vaccines

Messenger RNA (mRNA) has been identified as a potential platform for vaccine development against infectious diseases and have emerged as front runners in the unprecedented race to formulate an efficient vaccine for SARS-CoV-2, due to its capacity for rapid development and capability to operate potent adaptive immune responses [30,31]. The adaptability and quick creation of mRNA vaccines over traditional vaccinations such as live attenuated and inactivated virus and protein subunit vaccines are significant benefits [32]. Naturally transitory and cytosolically active messenger RNA (mRNA) molecules are being considered as a potentially safer and more powerful alternative to DNA for gene vaccination. When given naked, in liposomes, or coated on particles, optimized mRNA was proven to be a powerful gene vaccine carrier (Figure 1) [33]. Intramuscular injection of mRNA resulted in the local synthesis of an encoded protein and activation of immunological responses against an encoded antigen, demonstrating the usefulness of RNA in vaccination [34,35]. Exogenous mRNA has inherent immunostimulatory characteristics since it is recognized by a number of innate immune receptors found on the cell surface, endosomes, and cytoplasm. This has the potential to be helpful for vaccination because, in some instances, it may help to elicit powerful humoral and cellular immune responses by acting as an adjuvant [36]. RNA vaccines are based on the assumption that injected mRNA when snapped up by antigen presentation cells (APCs) and other target cells promotes the production of the correctly folded and glycosylated antigenic protein [37].

Because RNA activates endosomal and cytosolic RNA sensors upon cell entrance, these vaccines are self-adjuvanticity and produce both cellular and humoral immune responses to the encoded protein [30,38]. Two COVID-19 mRNA vaccines (mRNA-1273 as well as BNT162b2) were the first to be authorized and made accessible in the United States (by Pfizer-BioNTech and Moderna) [39]. Moderna developed mRNA-1273 vaccine is a lipid nanoparticle-encapsulated mRNA-based vaccine that replaces uridine with N1-methyl-pseudourine and encodes the SARS-CoV-2 full-length spike protein. The spike (S) protein, a class I fusion glycoprotein, is the major surface protein on the CoVvirion and the primary target for neutralizing antibodies. 2 proline substitutions (2P) at the apex of the central helix effectively stabilized MERS-CoV, SARS-CoV and HCoV-HKU1 S proteins. MERS-CoV-2 mRNA vaccination experiments, which found that full-length S-2P mRNA was more immunogenic than wild-type full-length S or secreted S-2P mRNA [40,41].The mRNA-1273 vaccine encodes the S-2P antigen, elicited both strong neutralizing antibody and CD8 T cell responses [42]. While BioNTech and Pfizer have developed BNT162b2 (Comirnaty®®), a nucleoside-modified mRNA vaccination for coronavirus disease prevention. BNT162b2 is made up of nucleoside-modified mRNA that has been synthesized in lipid nanoparticles (EMA) COVID-19 mRNA Vaccine).

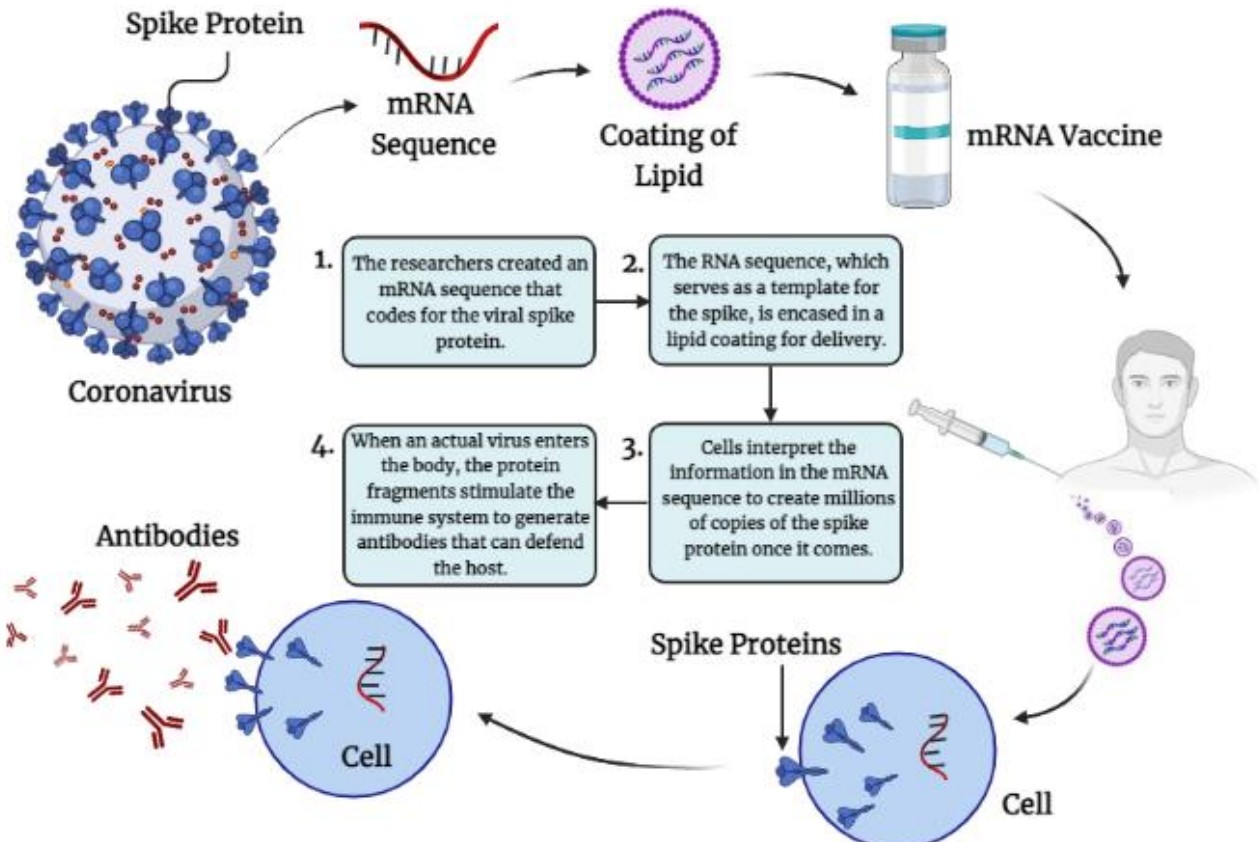

**Figure 1.** Shows how an RNA vaccine works against COVID-19.

The mRNA encodes the membrane-anchored, full-length SARS-CoV-19 spike protein and includes mutations that keep the spike protein in an antigenically favorable, prefusion shape. After intramuscular injection, the lipid nanoparticles shield the non-replicating RNA from degradation and allow it to be transported into host cells. Once within the host cell, the mRNA is translated into the SARS-CoV-2 spike protein, which is produced on the cell's surface. Transient production of this spike antigen generates neutralizing antibodies and cellular immunological responses, which may offer protection against COVID-19 [43,44]. Three additional m-RNA based vaccines are ongoing in phase Ior II clinical evaluation [45,46]. CVnCoV is a COVID-19 mRNA vaccine currently developed

by CureVacby utilizing non-chemically changed nucleotides in the mRNA; the vaccine is intended to generate robust and balanced immune response. K986P/V987P mutations with the RBDs preventing ACE2 from binding. CVnCoV encodes the whole SARS-CoV-2 S protein, which contains an intact S1/S2 cleavage site, as well as K986P and V987P alterations. In mice, CVnCoV elicited innate immune responses such as systemic IL-6 and IFN, indicating a pro-inflammatory milieu for the development of cellular and humoral immune responses [47,48]. Abogen developed ARCoV m-RNA based vaccine presently being studied in phase 1 clinical trials [49]. ARCoV is a modification of nucleoside Lipid Nanoparticle (LPs) encapsulated mRNA encoding the receptor-binding domain of the spike protein. In a preclinical investigation, ARCoV with a prime-boost vaccination schedule provided full protection for mice against a mouse-adapted SARS-CoV-2 strain and generated powerful neutralizing antibody and cellular responses in nonhuman primates [41]. Two doses of ARCoV vaccination in mice gave complete protection against a challenge with a SARS-CoV-2 mouse-adapted strain. Arcturus Therapeutics, Inc. is developing ARCT-021 is an mRNA vaccine that contains a self-replicating mRNA expressing SARS-CoV-2 spike protein encased in a lipid nanoparticle capable of inducing CD8+ cell-mediated and Th1/Th2-mediated immunity [47,50]. Because of their versatility, easy manufacturing method, and requirement of just pathogenic sequence for vaccine production, RNA-based vaccines are expected to be one of the quick answers to the pandemic issue. Furthermore, due to their intrinsic immunostimulatory characteristics, advanced self-amplifying and trans-amplifying RNA vaccine candidates enable powerful and persistent antigen synthesis in vivo at lower dosages [51].

### 4.1.2. Immunological Mechanisms of Different DNA Based Vaccine-Induced Protection against SARS-CoV-2

Findings in the early 1990s reported that plasmid DNA causes an immune response to the plasmid-encoded antigen, DNA vaccination has become the fastest expanding field in vaccine science [52,53]. DNA vaccines are composed of bacterial plasmids that, following in vivo injection, cell transfection and generate an encoded protein [54]. Genetic vaccines, with the exception of recombinant bacteria or viruses, are fully composed of DNA (as plasmids), which would be carried up by cells and transformed into protein. In gene-gun delivery, plasmid DNA is precipitated on with an inert particle (often beads of gold) and helium-blasted further into cells. Transfected cells then express the plasmid-encoded antigen, resulting in an immunological response. DNA viral vector-based vaccines (containing live or attenuated viruses), efficiently activate both MHC-II as well as MHC-I pathways, enabling an activation of CD4+ as well as CD8+ T cells [6]. DNA vaccines outperform several existing vaccinations based on recombinant proteins, recombinant viruses, or both [7,8,55]. Using DNA plasmids as a vector, DNA vaccines transmit genes or portions of genes encoding immunogenic antigens to host cells. As seen in Figure 2, this method efficiently generates both humoral and cell-mediated immune responses [56,57]. Furthermore, inherent plasmid DNA components such as CpGunmethylated regions might trigger innate immune responses, therefore boosting adaptive immune responses against the produced antigens. Although human clinical studies using DNA vaccines elicited both cellular and humoral responses, these responses are frequently insufficient to elicit meaningful therapeutic effects [57–59].

All DNA vaccines presently being investigated in clinical trials employ the S protein as the antigen. During preclinical testing, the INO-4800 vaccine-elicited both cellular and humoral immune responses in mice and guinea pigs within days of a single vaccination. As a result, the firm initiated a phase 1 open-label trial to assess INO-4800's safety, tolerability, and immunogenicity. INO-4800 vaccine is delivered intravenously using the CELLECTRA®®2000 device that generates a controlled electric field at the injection site to enhance the cellular uptake and expression of the DNA plasmid. Furthermore, preliminary research found that INO-4800 produced neutralized antibodies that inhibited the host from being bound to the SARS-CoV-2 spike protein receptor [42]. INO-4800 exhibited good safety and tolerability, as well as being immunogenic in 100% of vaccinated individuals,

generating either or both humoral and cellular immune responses. INO-4800 carries the plasmid pGX9501, which expresses a synthetic, optimized sequence of the SARS-CoV-2 full-length spike glycoprotein. Immunization by INO-4800 resulted in an immune response as measured by SARS-CoV-2 S1 + 2 protein binding IgG concentrations in sera [60]. bacTRL-Spike, COVID-19 DNA based vaccine developed by Symvivo, is presently in clinical trials. This bacTRL-Spike vaccine employs live, recombinant *Bifidobacterium longum* with synthetic plasmid DNA expressing SARS-CoV-2 S protein [42]. In order to activate an immune response via colonic lymphoid tissues, a DNA plasmid expressing trimeric S and a hybrid transporter protein in *Bifidobacterium longum* was administered to colonic epithelial cells [61]. GX-19 is a synthetic soluble SARS-CoV-2 spike (S) DNA-based vaccination candidate developed by Seo et al. In a dose-dependent way, GX-19 immunization produced not only S-specific systemic and pulmonary antibody responses, but also Th1-biased T cell responses in mice. Non-human primates immunized with GX-19 seroconverted quickly and there was a significant neutralizing antibody response as well as multifunctional CD8+ as well asCD4+T cell responses. GX-19 vaccination causes a rapid humoral response as well as Th1-biased immune responses in both mouse and nonhuman primate (NHP) models [62].

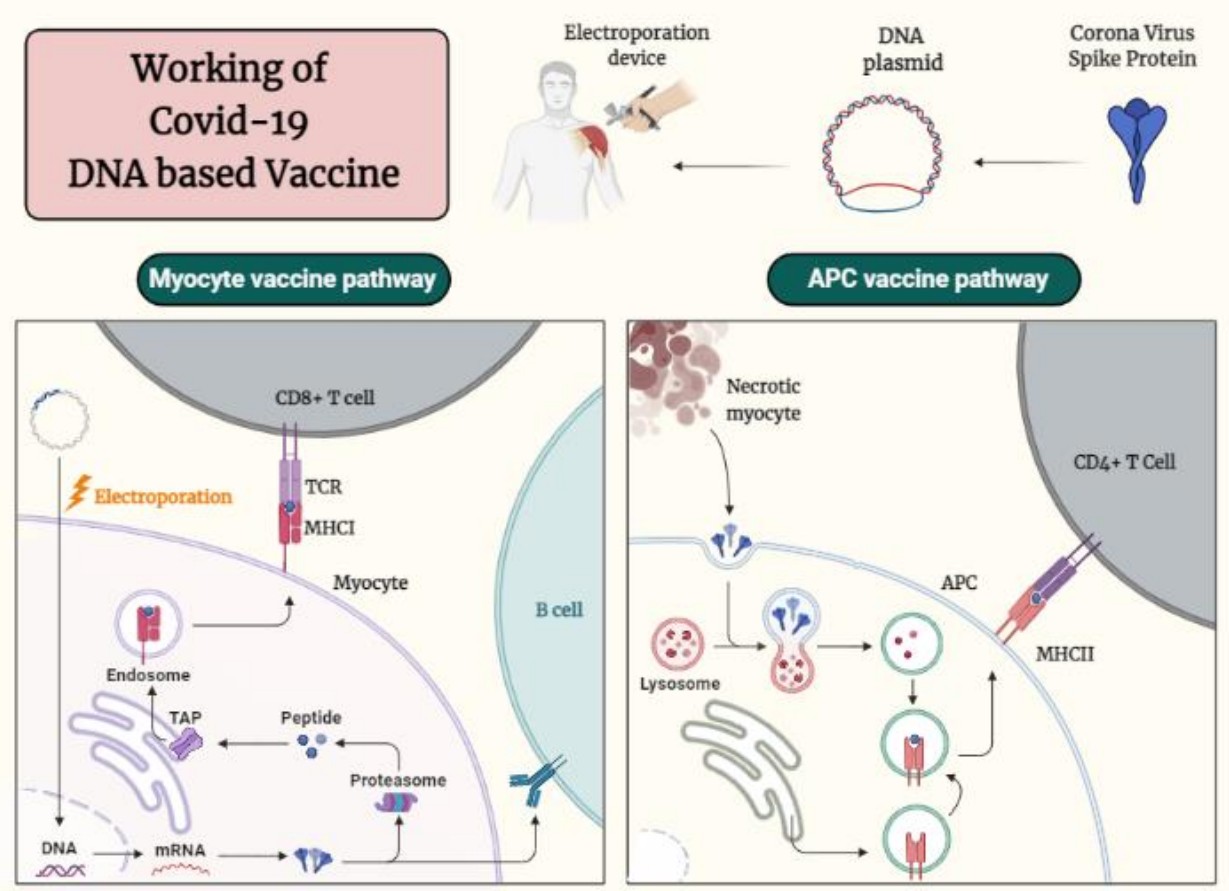

**Figure 2.** Shows the working of DNA Vaccine against COVID-19.

*4.2. Immunological Mechanisms of Different Adenoviral Vector-Based Vaccines-Induced Protection against SARS-CoV-2*

Adenoviral vectors were used with poxviral and DNA vectors to enhance immunogenicity, with eitheradenovirus or modified vaccinia virus Ankara prime-boost regimens improving both cellular and humoral responses [28]. Figure 3 presenting the mechanism that how viral vector-based vaccine works against SARS-CoV-2. Adenoviral vectors have been investigated as a platform for carrying and expressing a range of transgenes as a foundation for vaccine development [37]. The ChAdOx1 nCoV-19 adenoviral vector-based

vaccine (AZD1222) was constructed at Oxford University and consists of simian adenovirus vector ChAdOx1, which carries the full-pace structural surface glycoprotein (spike protein) of theSARS-CoV-2. ChAdOx1 nCoV-19 encodes a spike protein with a codon-optimized coding sequence [28,54,63]. ChAdOx1 nCoV-19 elicits a widespread and strong T cell response to both S antigen components. After vaccination, there was a significant increase in B cell activation and proliferation, and anti-IgA and IgG antibodies to the SARS-CoV-2 spike protein were easily identified in sera from vaccinated individuals [55]. Analyses of cytokine secretion after peptide stimulation of PBMCs revealed that IFN- and IL-2 production were higher in those who got the ChAdOx1 vaccination compared to controls, while IL-4 and IL-13 levels were not. Similarly, flow cytometry phenotyping revealed that CD4+ T cells produced primarily Th1 cytokines (IFN-, IL-2, and TNF-) rather than Th2 cytokines (IL-5 and IL-13). Importantly, it showed that immunization with ChAdOx1 nCoV-19 generates a mainly Th1 response using a variety of methods (multiplex cytokine profiling, ICS analysis, and antibody isotype profiling). In a phase 1/2 research, a single dosage of ChAdOx1 nCoV-19 resulted in a rise in spike-specific antibodies by day 28 and neutralizing antibodies in all participants after a booster dose. After vaccination, there was a significant increase in B cell activation and proliferation, and anti-IgA and IgG antibodies to the SARS-CoV-2 spike protein were easily identified in sera from vaccinated participants [28]. T-cell responses believed to play an important role in COVID-19 mitigation; persons who have been treated but asymptomatic developed a robust memory T-cell response in the absenteeism of clinical disease, despite the lack of a recognizable humoral response [29,52]. ChAdOx1 nCoV-19 was shown to be safe, tolerable, and immunogenic, with reactogenicity decreased by paracetamol. A sole dose elicited both humoral and cellular responses against SARS-CoV-2, and just a booster dose increased neutralizing antibody titers [28,53].

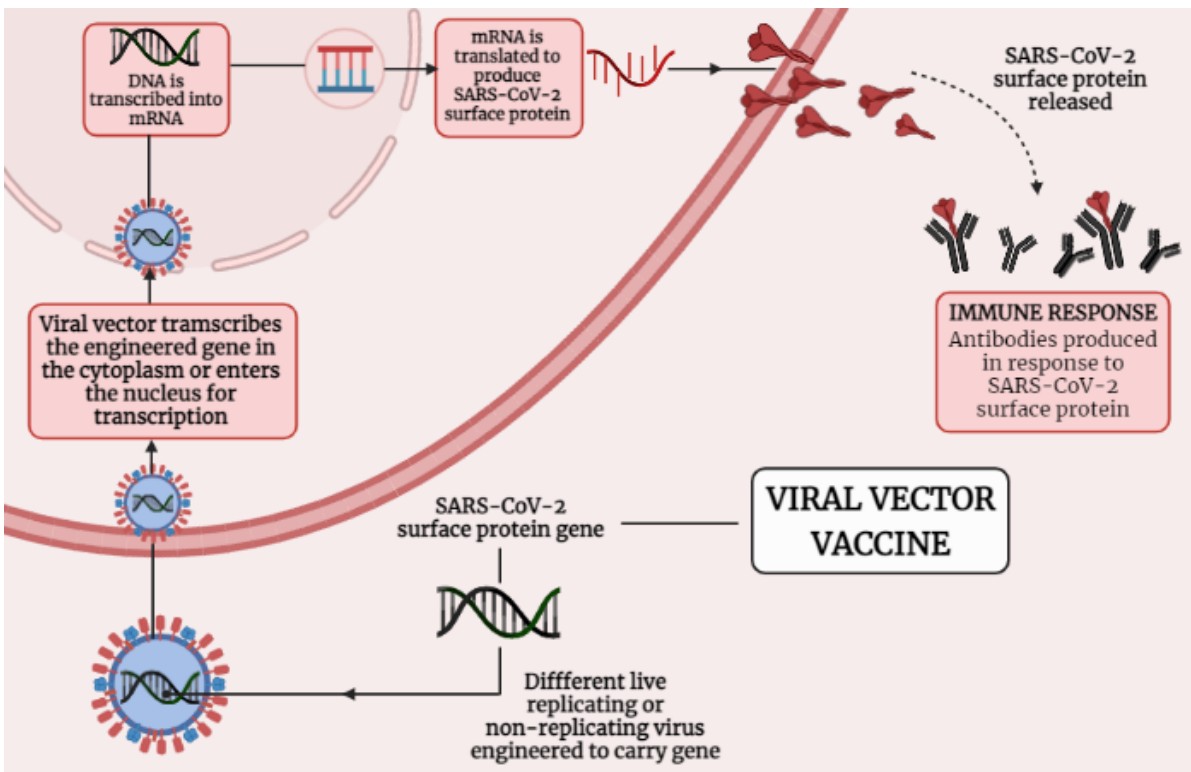

**Figure 3.** Shows the mechanism of Viral vector-based vaccine against SARS-CoV-2.

Gam-COVID-Vac is indeed a combined vector vaccine based on rAd type 5 (recombinant adenovirus type 5, rAd5) as well asrAd type 26 (recombinant adenovirus type 26, rAd26), which all include the SARS-CoV-2 full-spanglycoprotein S gene (rAd5-S as well as rAd26-S). rAd26-S and rAd5-S are given intramuscularly individually at a 21-day

interval. Recipients produced strong antibody responses to the spike protein, including neutralizing antibodies, which are a subset of total immunoglobulin that prevents the virus from attaching to its receptor. They also demonstrated T-cell responses, which is consistent with an immunological response that should not fade rapidly [56–58]. Sputnik V is efficient against novel coronavirus strains. The vaccine induces protective neutralizing antibody titers against new strains including Beta B.1.351 (initially defined in South Africa), Alpha B.1.1.7 (first identified in the United Kingdom), Delta B.1.617.2, and B.1.617.3 (initially recognized in India) as well as Gamma P.1 (initially indicated in Brazil), and variants B.1.1.141 as well asB.1.1.317 with receptor-binding domain (RBD) mutations identified in Moscow [33,34].The vaccine's phase 1/2 clinical studies were finished in August 2020. The vaccination was well handled and highly immunogenic in healthy individuals, according to the findings. As a consequence, the vaccine candidate was granted provisional approval in Russia under national regulations. It provides early effectiveness and safety data from a phase 3 multicenter study of Gam-COVID-Vac in adults, including a sub analysis of individuals over 60 years old [56,57]. In all age groups, the vaccination elicited strong humoral (n = 342) and cellular (n = 44) immunological responses. This interim analysis of the Gam-COVID-Vac phase 3 study found that it was 91.6 percent effective against COVID-19 and well-tolerated in a large cohort [57]. Earlier phase 1/2 data, released in September 2020, shown excellent safety results and indicated that the immune response was at a level compatible with protection 10. The Russian Federation's Ministry of Health's Gamaleya National Research Center of Epidemiology and Microbiology as well as the Russian Direct Investment Fund (RDIF, Russia's sovereign wealth fund) confirmed recently that the Sputnik V vaccine illustrated effectiveness of 97.6 percent, predicated on a research observational study on coronavirus rates of infection among Russians vaccinated with both constituents [60]. SARS-CoV-2-specific antibody responses in healthcare professionals in Argentina following Sputnik V vaccination, with IgG anti-spike titers and neutralizing capability measured after one and two doses in a sample of naive or already infected volunteers. 94 percent of naive individuals produce spike-specific IgG antibodies 21 days after receiving the first dose of vaccination. A total of 94% of vaccinations generated specific anti-spike antibody responses, with 90% exhibiting WT virus-neutralizing capability. Significantly, among seropositive individuals, a single dose of Sputnik V vaccination elicited a rapid and strong immune response, with neutralizing titers that surpassed those seen in seronegative participants who received two doses. The significant seroconversion rate following a single dose in naive individuals implies that postponing second dose delivery may be beneficial in increasing the number of persons immunized [35]. Ad26.COV2.S is a replication-incompetent and recombinant adenovirus serotype 26 (Ad26) vector based vaccine encoding a full-length and stabilized SARS-CoV-2 spike (S) protein [6]. Vaccines relied on Ad26 seems to be typically reliable as well as extremely immunogenic [64–66]. COV1001 is a placebo controlled, multicenter, double-blind, randomized phase 1–2a clinical trial that will recruit healthy adults in two age cohorts to evaluate the reactogenicity, safety, as well asimmunogenicity of Ad26. COV2.S [55]. Poorly neutralizing humoral immunity and Th2-skewed cellular immune responses have been linked to a potential risk of vaccine-associated increased respiratory illness (VAERD). In one investigation, all elicited CD4+ T-cell responses to Ad26.COV2.S were Th1-biased, which is consistent with previous findings using the Ad26-based vaccination platform [34,52,53]. The accompanying prolonged CD8+ T-cell responses (though somewhat at lower elevations in elderly people than that in teenagers) and strong humoral responses diminish the potential risk of VAERD substantially [55]. After vaccination with Ad26-based vaccine encoding S.PP (Ad26.S.PP), the IFN- level to IL-4, IL-10, or IL-5concentration ratios were high, indicating that the response was Th1-skewed. Ad26.S.PP produced a dominant Th1 response in BALB/c mice when combined with high Nab titers, lowering the potential risk of vaccine-associated increased illness. The effectiveness of Ad26.S.PP vaccine elicits protective immunity against SARS-CoV-2 infection was demonstrated in a non-human primate challenge paradigm [67–69].

### 5. Vaccines and Its Role in Inducing Humoral Adaptive Immunity

Recent research has revealed that the novel SARS-CoV-2 virus employs a similar mechanism for cell entrance [70]. To connect to host cells, the viral S protein attaches to the angiotensin-converting enzyme 2 (ACE2), the viral receptor. The S protein is then primed by host cell proteases, furin, and the serine proteases TMPRSS2 and TMPRSS4, allowing viral and cellular membranes to fuse and viral RNA to enter the host cell [71]. Long-term protective immunity is supplied by vaccination antigen (or pathogen)-specific immune effectors and the activation of immunological memory cells that can be effectively and quickly reactivated in the event of pathogen exposure [72–74]. Most vaccines that have been approved thus far stimulate antibodies generated by B cells, which are believed to be responsible for the vaccine's long-term protection [75,76]. Vaccine antigen and pathogen binding to B cell receptors (antibody in membrane-bound form) induces the production of an initial activation marker CD69 and also a chemokine receptor CCR7 that drives antigen-specific B cells onto secondary lymphoid tissue T cell zones [77–80]. Vaccine antigen-specific B cells are likely to engage with newly activated T cells and DCs, particularly follicular DCs with specific surface molecules, at this site (CD40, CD80, CD86). This T cell assistance accelerates B cell development into antibody-secreting, short-lived plasma cells that generate low-affinity germ-line encoded antibodies [80]. The development of neutralizing antibodies aimed towards spike protein is a key component of successful vaccination. This is the foundation for many clinical trials, as well as the creation of monoclonal antibody cocktails, which have proven crucial in COVID-19 treatments in the absence of a vaccine. However, nothing was known until recently about what constituted a successful immunological response to COVID-19. The architecture of antibodies targeting the SARS-CoV-2 receptor-binding domain (RBD) is an essential factor. Recent studies have demonstrated that antibodies bound to the receptor-binding domain RBD are important for long-term protective immunity against COVID-19 infection and are linked with improved patient survival [81–84]. Virus-neutralizing antibodies are largely responsible for the protection provided by presently available vaccinations. These antibodies often inhibit the virus's contact with its cellular receptor or prevent the virus from undergoing the conformational changes necessary for fusion with the cell membrane. Vaccination's objective is to generate long-term protective immunity, which is a feature of adaptive immunity.

### 6. Conclusions

The arrival of the SARS-CoV-2 outbreak has resulted in the world economy's demise. Researchers from all across the globe are working together to slow the rapidly increasing pace of COVID-19. Several vaccination systems have been used concurrently, with concurrent pre-clinical and clinical stages. This has also sped up the production process; nevertheless, there is still room for improvement. The accomplishments in creating COVID-19 vaccines are enormous and unmatched in medical history. Never before have novel vaccine technologies been put into practice so quickly, nor have manufacturing capacity for billions of vaccine doses been developed so efficiently from the ground up. Also, with the present condition of COVID-19 instances, it appears that the vaccination testing and its development will carry on for the foreseeable future until more and more vaccines are authorized for urgent use. Still, there seems to be an ongoing debate over vaccine-induced herd immunity or whether vaccinated persons may still spread the virus while infected, even if they are shielded from producing illness. Despite the fact that a large number of COVID-19 vaccines have already been approved in several countries and are being used in mass vaccination programs, there is still a huge list of candidate vaccines in the pipeline. According to WHO, the figure is far greater than 200, including 63 in clinical development and 174 in preclinical development. Upgrades are always conceivable, and interesting techniques are available. Next-generation candidates, on the other hand, will confront the challenge of showing vaccine effectiveness in an environment where cognitively prepared vaccines still exist. Placebo controls (as used in phase 3 trials of presently authorized vaccinations) will be impossible to implement in such a circumstance and demonstrating

non-inferiority to existent vaccines would necessitate much more participants, increasing the cost of studies. A solution to such challenges would have been the development of a trustworthy in vitro correlation of protection that is acceptable to licensing authorities. In conclusion, we may be pleased with present progress and sure that new difficulties will be met with the same zeal and inventiveness that was necessary for previous achievements.

**Author Contributions:** P.T., T.K.U., H.G.: conceptualized and ideated the scheme, K.G., H.G., P.B., A.T., A.D. (Aman Dixit), A.D. (Abhijit Dey): writing—performed the literature survey and wrote initial draft, K.G., H.G.: performed artwork and drafted the manuscript, K.G., H.G., P.B., A.T., A.D. (Aman Dixit), A.D. (Abhijit Dey), A.K.P., D.K., S.R., K.K.K., P.K.G., M.B.: reviewed, writing and editing the manuscript, P.T. and T.K.U.: Supervision and approved the final manuscript. N.K.J. and K.K.K. reviewing and supervision, F.K. and P.P. helped in grammar correction, editing and formatting. All authors have read and agreed to the published version of the manuscript.

**Funding:** This research received no external funding.

**Institutional Review Board Statement:** This is not applicable for the present manuscript.

**Informed Consent Statement:** This is not applicable for the present manuscript.

**Data Availability Statement:** This is not applicable for the present manuscript. The data reviewed herein is from previously published manuscripts and well referred appropriately.

**Conflicts of Interest:** The authors declare no conflict of interest.

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
