# Peer review of "Immunological Mechanisms of Vaccine-Induced Protection against SARS-CoV-2 in Humans"

_2673-5601, doi:10.3390/immuno1040032_

Round 1

Reviewer 1 Report

The authors present a well organized and comprehensive review of the current status of COVID-19 vaccines across the globe. I have no major or minor comments but to encourage another round of editing for clarity, for example the use of repertory in line 64, is correct but not common in science writing and repertoire would be more easily understood by a wider audience. In addition there were a couple places where I could tell this wasn't written by a native English speaker. Which is ok, and nothing was grammatically incorrect, but the readers experience could be improved with just a few changes. Otherwise a very nicely done review. 

Author Response

Comment: The authors present a well-organized and comprehensive review of the current status of COVID-19 vaccines across the globe. I have no major or minor comments but to encourage another round of editing for clarity, for example the use of repertory in line 64, is correct but not common in science writing and repertoire would be more easily understood by a wider audience. In addition there were a couple places where I could tell this wasn't written by a native English speaker. Which is ok, and nothing was grammatically incorrect, but the readers experience could be improved with just a few changes. Otherwise a very nicely done review. 

Response: Organization and English language has been modified now from the English native speaker and manuscript has been modified accordingly. All suggested suggestions were also laid down in this manuscript. Thank you for constructive suggestions which improves the content of this manuscript.

Reviewer 2 Report

In this manuscript the authors present a review of the currently administered SARS-CoV-2 vaccines and the ones evaluated in clinical trials. The list of vaccines described is accurate and actualized with information not only about their efficacy but also including the generated immune responses that contribute to their observed effects. However, the data presented does not reflect the title of the manuscript (Vaccine induced immune responses against SARS-Cov-2 infections) as the immune responses that are triggered are described in a very superficial manner. Moreover, a more concise and focused introduction of the most important immunity players that are the basis for the mechanism of action is necessary. While the antibody response is properly explained to highlight its importance in protection from SARS-CoV infection, the T cell response  (CD8 and CD4) requires better introduction to understand the long term immunity offered by the different vaccination regimes.

I acknowledge that this is an evolving research field but there is already significant data regarding the durability and long term protection offered by the different vaccines. Therefore, some of the newest findings in this area should be mentioned in a potential revised form of the manuscript.

Overall the paragraphs are somewhat disorganized without the description of new acronyms that makes readability difficult. Moreover there are significant typos and there is often poor wording choice.

Minor issues:

Page 3

“Covishield is a SARS-CoV-2 spike protein-based vaccine that is 70.42 percent 79 effective after two doses. Covaxin, on the other hand, is an inactivated vaccination of 80 SARS-CoV-2 that is 60% effective..”

Effective at what?

TABLE:  please define DGCI.

Page 5

“With exception of innate immune responses, adaptive immune responses are extremely precise to the pathogen that elicited them. They can however offer long-term resistance”.

                  As it is written, it is not clear. Please, rewrite.

“The synchronization of T and B cell immune responses to the SARS CoV-2 virus is referred to as adaptive immunity”.

Actually, the synchronization of T and B cell immune responses is referred to as adaptive immunity, regardless of the pathogen.

“Adaptive immunity involves the coordination of T and B cell immune responses to the SARS CoV-2 virus”.

                  This sentence is a repetition.

“This is due to the ability of B memory cells and long-lived plasma cells in the bone marrow to reactivate antigen-specific responses to the SARS-CoV-2 RBD if exposed again.

                  Not only responses to RBD, but also many other targets.

“Moreover, this ignores the relevance of T cell memory for COVID-19 antigenic determinates, which can lead to efficient cytotoxic T cell immunity and aid in B cell responses.

                  Not clear what “ignores” means in this sentence.

Paragraph 4.1 starts with an introduction that has a reference published in 1995. Many newer reviews have been published. Please, select a more updated one.

Page 6

“An immune response is  instead largely initiated by specialized bone marrow (BM)-derived antigen presentation cells (APC) known as dendritic cells (DC). The importance of bone marrow-derived DC has been shown largely through the use of BM-reconstituted chimeras.

APC include dendritic cells, B cells, macrophages and Langerhans cells. There’s no need to emphasize on BM-dendritic cells at this point.

“Several vaccinations have been produced against SARS-CoV-2, the virus that causes COVID-19”

                  At this point, there’s no need to repeat this.

“full-length S-2P mRNA was more immunogenic than wild-type full-length S or secreted S-2P mRNA”

                  Briefly describe S-2P.

Page 7

“K986P and V987P alterations”

            If these alterations are mentioned, they should be briefly described.

“LNP-encapsulate”

            Please describe LNP

Page 8

“DNA vaccines seem to be bacterial plasmids”

Either they are or they are not. Please rewrite

“Genetic vaccines, with the exception of recombinant bacteria or viruses, are fully composed of RNA (as mRNA) or DNA (as plasmids)”

                  Not clear why mRNA is part of this section, supposedly in a DNA-based vaccines section.

“DNA vaccines, such as live or attenuated viruses, efficiently activate both MHC-II as well as MHC-I pathways, enabling an activation of CD4+ as well as CD8+ T cells [52]. DNA vaccines outperform several existing vaccinations based on recombinant proteins, recombinant viruses, or both.”

                  It is a bit confusing as it is not clear what is plasmid-based versus viral-vector-based. Please, rewrite.

The paragraph describing INO-4800 is a bit disorganized. Also, it is not clear how intravenous electroporation is carried-out(?)

“INO-4800 produced neutralized antibodies which prevented whether the host is bound by the SARS-CoV-2 S protein. receptor ACE2”

                  Please rewrite.

Page 9

“Seo and his colleagues”

            Please replace by Seo et al.

“GX-19 vaccination induces a contemporaneous humoral response”

                  Not clear what the meaning is.

Please, find alternative references for 25

The entire paragraph of ChAdOx1 is very disorganized, please rewrite.

Page 10

“including neutralizing antibodies, which are a subset of total immunoglobulin that prevents the virus from attaching to its receptor”

                  This concept should have been introduced earlier in the manuscript

Page 11

                  Please define Ad26.S.PP.

Author Response

Comment: In this manuscript the authors present a review of the currently administered SARS-CoV-2 vaccines and the ones evaluated in clinical trials. The list of vaccines described is accurate and actualized with information not only about their efficacy but also including the generated immune responses that contribute to their observed effects. However, the data presented does not reflect the title of the manuscript (Vaccine induced immune responses against SARS-Cov-2 infections) as the immune responses that are triggered are described in a very superficial manner. Moreover, a more concise and focused introduction of the most important immunity players that are the basis for the mechanism of action is necessary. While the antibody response is properly explained to highlight its importance in protection from SARS-CoV infection, the T cell response (CD8 and CD4) requires better introduction to understand the long term immunity offered by the different vaccination regimes.

I acknowledge that this is an evolving research field but there is already significant data regarding the durability and long-term protection offered by the different vaccines. Therefore, some of the newest findings in this area should be mentioned in a potential revised form of the manuscript.

Overall the paragraphs are somewhat disorganized without the description of new acronyms that makes readability difficult. Moreover, there are significant typos and there is often poor wording choice.

Response: We would like to thank you for your valuable comments. The Title of this manuscript has been modified now. Introduction has also been modified we introduce all suggestion in our introduction section including antibody response, T cell response offered by different vaccine regime is properly explained now with its importance in protection from SARS-CoV infection. Along with this some of the newest findings regarding long term protection offered by different vaccines has been incorporate in vaccine section 4. Organization of paragraph has been modified with acronyms. All suggested suggestions were also laid down in this manuscript and changes highlighted in with yellow shade. Thank you for constructive suggestions which improves the content of this manuscript.

Comment: Page 3

“Covishield is a SARS-CoV-2 spike protein-based vaccine that is 70.42 percent 79 effective after two doses. Covaxin, on the other hand, is an inactivated vaccination of 80 SARS-CoV-2 that is 60% effective..”

Effective at what?

Response: It has been modified and highlighted with yellow shade section 1 (Page 3).

Comment: TABLE:  please define DGCI.

Response:It has been modified and highlighted with yellow shade in table 1.

Comment: Page 5

“With exception of innate immune responses, adaptive immune responses are extremely precise to the pathogen that elicited them. They can however offer long-term resistance”.

As it is written, it is not clear. Please, rewrite.

Response: It has been modified now and highlighted with yellow shade in section 2. Page no 5.

Comment: “The synchronization of T and B cell immune responses to the SARS CoV-2 virus is referred to as adaptive immunity”.

Actually, the synchronization of T and B cell immune responses is referred to as adaptive immunity, regardless of the pathogen.

“Adaptive immunity involves the coordination of T and B cell immune responses to the SARS CoV-2 virus”.

This sentence is a repetition.

Response: It has been modified and highlighted with yellow shade in section 2. Page no 5.

Comment: “This is due to the ability of B memory cells and long-lived plasma cells in the bone marrow to reactivate antigen-specific responses to the SARS-CoV-2 RBD if exposed again.

Not only responses to RBD, but also many other targets.

Response:It has been modified and highlighted with yellow shade in section 3. Page no 5.

Comment: “Moreover, this ignores the relevance of T cell memory for COVID-19 antigenic determinates, which can lead to efficient cytotoxic T cell immunity and aid in B cell responses.

Not clear what “ignores” means in this sentence.

Response:It has been modified and highlighted with yellow shade in section 3. Page no 5.

Comment: Paragraph 4.1 starts with an introduction that has a reference published in 1995. Many newer reviews have been published. Please, select a more updated one.

Response: It has been modified and highlighted with yellow shade in reference section.

Comment: Page 6

“An immune response is  instead largely initiated by specialized bone marrow (BM)-derived antigen presentation cells (APC) known as dendritic cells (DC). The importance of bone marrow-derived DC has been shown largely through the use of BM-reconstituted chimeras.

APC include dendritic cells, B cells, macrophages and Langerhans cells. There’s no need to emphasize on BM-dendritic cells at this point.

Response: It has been modified and highlighted with yellow shade in section 4.1 Page no 6.

Comment: “Several vaccinations have been produced against SARS-CoV-2, the virus that causes COVID-19”

At this point, there’s no need to repeat this.

Response:This point has been removed from this manuscript.

Comment: “full-length S-2P mRNA was more immunogenic than wild-type full-length S or secreted S-2P mRNA”

Briefly describe S-2P.

Response: It has been described in section 4.1.1 Page no 6 and highlighted with yellow shade.

Comment: Page 7

“K986P and V987P alterations”

If these alterations are mentioned, they should be briefly described.

Response:It has been described in section 4.1.1 Page no 7 and highlighted with yellow shade.

Comment: “LNP-encapsulate”

Please describe LNP

Response: It has been described in section 4.1.1 Page no 7 and highlighted with yellow shade.

Comment: “DNA vaccines seem to be bacterial plasmids”

Either they are or they are not. Please rewrite

Response: It has been modified in section 4.1.2 Page no 8 and highlighted with yellow shade.

Comment: “Genetic vaccines, with the exception of recombinant bacteria or viruses, are fully composed of RNA (as mRNA) or DNA (as plasmids)”

 Not clear why mRNA is part of this section, supposedly in a DNA-based vaccines section.

Response: It has been modified in section 4.1.2 Page no 8 and highlighted with yellow shade.

Comment: “DNA vaccines, such as live or attenuated viruses, efficiently activate both MHC-II as well as MHC-I pathways, enabling an activation of CD4+ as well as CD8+ T cells [52]. DNA vaccines outperform several existing vaccinations based on recombinant proteins, recombinant viruses, or both.”

It is a bit confusing as it is not clear what is plasmid-based versus viral-vector-based. Please, rewrite.

Response: It has been modified in section 4.1.2 Page no 8 and highlighted with yellow shade.

Comment: The paragraph describing INO-4800 is a bit disorganized. Also, it is not clear how intravenous electroporation is carried-out(?)

Response: It has been modified in section 4.1.2 Page no 8 and highlighted with yellow shade.

Comment: “INO-4800 produced neutralized antibodies which prevented whether the host is bound by the SARS-CoV-2 S protein. receptor ACE2”

Please rewrite.

Response: It has been modified in section 4.1.2 Page no 8 and highlighted with yellow shade.

Comment: Page 9

“Seo and his colleagues”

Please replace by Seo et al.

Response: It has been modified in section 4.1.2 Page no 9 and highlighted with yellow shade.

Comment: “GX-19 vaccination induces a contemporaneous humoral response”

 Not clear what the meaning is.

Response: It has been modified in section 4.1.2 Page no 9 and highlighted with yellow shade.

Comment: Please, find alternative references for 25

The entire paragraph of ChAdOx1 is very disorganized, please rewrite.

Response: It has been modified and highlighted with yellow shade in reference section.

Comment: Page 10

“including neutralizing antibodies, which are a subset of total immunoglobulin that prevents the virus from attaching to its receptor”

This concept should have been introduced earlier in the manuscript

Response: It has been modified and highlighted with yellow shade in section1. Page 3.

Round 2

Reviewer 2 Report

The review looks fine but still needs grammatical corrections.